# Protective Vaccination Used by Doctors for Prevention of Infections

**DOI:** 10.3390/ijerph20054153

**Published:** 2023-02-25

**Authors:** Beata Zastawna, Roman Załuska, Anna Milewska, Agnieszka Zdęba-Mozoła, Agnieszka Ogonowska, Remigiusz Kozłowski, Anna Owczarek, Michał Marczak

**Affiliations:** 1Department of Management and Logistics in Health Care, Medical University of Lodz, 90-131 Lodz, Poland; 2Department of Statistics and Medical Informatics, Medical University of Bialystok, 15-089 Bialystok, Poland; 3Center for Security Technologies in Logistics, Faculty of Management, University of Lodz, 90-237 Lodz, Poland; 4Collegium of Management, WSB University in Warsaw, 03-204 Warszawa, Poland

**Keywords:** vaccination, immunization, vaccine preventable diseases, healthcare workers, vaccination hesitancy, prophylactic methods

## Abstract

Doctors, as with all healthcare workers, are a specific risk group due to a high probability of contact with contagious pathogens. An online survey was conducted among Polish doctors to establish their use of protective vaccination to decrease their personal risk of infection. The online survey was conducted using questions about medics’ vaccination decisions and approaches. The results revealed that immunization against VPDs for most participants was not adequate based on recommendations or developments in vaccinology. To increase vaccination as a prophylactic method among doctors, especially those not involved in the immunization of patients, an educational campaign is demanded. As non-immunized medics are at risk themselves and are also a threat to the safety of patients, legal changes and the monitoring of vaccine acceptance and perception among medics are required.

## 1. Introduction

The pandemic of a new, virulent coronavirus variant, SARS-CoV-2, has resulted in an unprecedented need for the use of preventive methods, particularly in healthcare. At the same time, reflections arose in the medical community concerning the current degree to which procedures were followed and to which doctors made use of existing methods for preventing infectious diseases. The threat of transmitting pathogens is the main occupational hazard for persons working with contagious patients. The danger of infection applies not only to them, but also to their close family, colleagues and treated patients [1,2,3,4,5].

The most effective method to reduce the probability of infection is the immunization of medical employees against Vaccine Preventable Diseases (VPDs) [1,4,6]. Vaccines are usually performed in a planned manner, as a prophylaxis, providing with resistance to potential infection from contact with a specific pathogen. In the case of some diseases, post-exposure vaccination is possible, e.g., with measles or varicella vaccine within 72 h after direct contact with an ill person [6,7,8,9,10].

The development of vaccinology has been systematically expanding the possibility of using the protection afforded by vaccinations. Changes have included introducing new vaccines, manufacturing with the use of more and more advanced technologies, and also extending indications and changing their administration schedules. There are also new recommendations for boosters for specific risk groups, including health care workers [1,2,3,4,5,6,7,8,11,12,13,14]. Multiple studies indicating that the employees themselves are a frequent source of hospital outbreaks, and that the morbidity and mortality of older patients with VPDs depend strictly on the level of immunity of the employees are also additional arguments for the vaccination of medical personnel [2,3,4,5].

The annually updated Protective Vaccination Programme in Poland provides infants and children up to the age of 18 with free immunization according to WHO recommendations and due to epidemiological and economic situation in Poland. The vaccination schedule includes protection against tuberculosis, diphtheria, pertussis and tetanus. There are also protective vaccinations, introduced in the following years as mandatory, against poliomyelitis (since 1972), mumps (1 dose since 1975, 2 doses since 1991), rubella (1988, the last two vaccines replaced by MMR vaccines in 2004), hepatitis B (1996), Haemophilus Influenzae (2007), pneumococci (2017), rotaviruses (2021) and Human Papilloma Virus (HPV) (2023). The optional vaccinations not financed by state are against varicella, hepatitis A, meningococci B and ACWY and tick-borne encephalitis [7,8].

The Protective Vaccination Programme also contains recommendations for all adults, including selected risk groups [7,8]. According to this program, vaccinations against hepatitis B have been free and obligatory for some years for all medics in Poland, and they are also financed for students in medical schools and of medical universities [12]. Moreover, annual vaccinations against the flu are recommended for all healthcare employees, and a booster vaccination against pertussis every 10 years is recommended for personnel working with infants and children [1,4,7,11]. Measles vaccination is recommended for all adults who are not immune—i.e., without two documented doses or having had measles in childhood [6,7,10]. The Protective Vaccinations Programme also indicates the need to protect medical personnel and laboratory employees against meningococcal infections. Regardless of the place of work and profession, non-vaccinated young people should get immunized against rubella. Pregnant women are advised to get vaccinated against influenza and boosted against pertussis in the third trimester of pregnancy. Persons older than 50 are recommended to vaccinate themselves against pneumococcal disease. Indications for the use of protective immunizations in adults with polymorbidity are also being extended [1,6,7].

The preventive vaccination of medical personnel is certainly advantageous for employers, since by reducing the risk of transmission of contagious factors, it contributes to reducing the amount of illness and morbidity among employees and patients [2,3,6,9,13]. A decreased personnel absence due to sickness during periods of frequent infections makes the organization of work easier.

The principles of the professional conduct of a medical doctor in Poland are specified in the Medical Code of Ethics and in the appropriate legal regulations. The primary duties established therein include the prevention of illnesses, following the current medical knowledge and promoting health-oriented attitudes. There is however no obligation to get the recommended vaccinations, nor is there a legal ability to monitor the degree to which the Protective Vaccination Programme is followed among the medical employees. The doctors’ knowledge concerning current vaccine recommendations is not evaluated in this study.

The goal of the study was to learn the attitudes of medical personnel towards the available protective vaccinations.

## 2. Materials and Methods

This study was conducted on doctors employed in hospitals at varying levels working in three areas of Poland. An original internet questionnaire was used. A total of 182 completed questionnaires were obtained; 38% of doctors were up to 50 years old, 52% were in the age range of 51 to 65 years, and 9% were older than 65 years. Women made up 73.5%.(Table 1). Non-surgical specialties dominated (Figure 1).

Since some doctors had more than one specialty, the following categories (most frequently occurring) were created for the analysis: only internal diseases (*n* = 34, 18.8%); only pediatrics (*n* = 14, 7.7%); only family medicine (*n* = 47; 26.0%); pediatrics and family medicine (*n* = 13; 7.2%); internal diseases and family medicine (*n* = 55; 30.4%).

For statistical analysis, a Chi-square test of independence was used in order to check for dependencies between qualitative characteristics. A correspondence analysis was used in order to present relationships between qualitative characteristics. Aggregations and closest distances of the categories we were interested in were analyzed for the interpretation of correspondence maps. Results at a level of *p* < 0.05 were deemed to be statistically significant. The Statistica 13.03 package was used for the calculations.

## 3. Results

For the additional vaccinations, the largest number of doctors had vaccinated themselves as adults against hepatitis B (79%), then against influenza (77%), and a much lower number with other available vaccines (Figure 2); 11% of respondents were only vaccinated with their obligatory childhood vaccinations, in accordance with the vaccination schedule.

The additional vaccinations were not significantly statistically dependent on gender, place of work, or specialization (Table 1)There was only a dependency between age and hepatitis B vaccination (*p* = 0.004) and between pertussis booster vaccination (*p* < 0.001).

The doctors that were asked about the knowledge of their own immunity to the measles virus mostly declared that they were immune to the measles virus (43% had been ill with measles, 33% were vaccinated with two doses). A total of 8% were vaccinated with a single dose, almost 2% were not vaccinated nor were ever infected with measles, and 15% had no knowledge about their immunity.

Almost half of those surveyed declared regular vaccinations against influenza. One third of the doctors vaccinate themselves irregularly, whereas almost 20% said they do not vaccinate themselves. There is no statistically significant dependency between influenza vaccination and the doctor’s age and gender (Table 1).

A correspondence analysis allowed for observing a relationship between influenza vaccination and the doctor’s specialization (Figure 3). Aggregations present the most typical attitudes. Aggregation A: 26% of internal disease specialists do not vaccinate themselves, and 23% of doctors with two specializations (internal and family medicine) do not vaccinate themselves. Aggregation B: 70% of family doctors vaccinate themselves regularly. Aggregation C: 36% of pediatric doctors vaccinate themselves but not regularly, and 33% of doctors with a double specialization (pediatric and family medicine) do vaccinate themselves, but irregularly.

The surveyed were asked whether they recommend vaccination against human papillomavirus (HPV) to their own children or to children of close acquaintances. It turned out that most frequently it was recommended only to girls (45%), 39% of the doctors recommended this vaccination to children of both sexes, and 16% of the doctors did not recommend it to related children at all.

A correspondence analysis demonstrated a relationship between age and recommendation of HPV vaccination. The younger the doctors were, the wider the group of children they recommended vaccination to. Aggregation A—doctors aged up to 50 years recommended to children of both sexes, aggregation B—doctors aged 51–65 recommended only to girls, aggregation C—doctors aged over 65 did not recommend this vaccination (Figure 4).

The relationship, demonstrated on a correspondence map, between the specialization and the recommendation of HPV vaccination to own children or children of close family is presented on Figure 5. The aggregation visible on the figure demonstrates a relationship between specialization in internal diseases and recommending of vaccinations only to girls—this only applied to 55% of doctors. Also, for experts in both pediatrics and family medicine, the most typical recommendation was “yes, to girls” (69% of doctors), whereas vaccinations to both girls and boys were recommended by 71% of pediatricians and 44% of doctors with two specializations (internal diseases and family medicine).

According to those surveyed, the highest impact on the increase of vaccination of doctors would result from the co-financing of protective vaccinations for employees by the work establishment (76%), the ability to get vaccinated at their place of work and during the time of work (61%), obtaining information on the increase of incidence of a specific disease in the nearest vicinity (33%), and recommendation of vaccination issued by the work establishment’s management (29%). More rarely it was pointed out that a positive impact on vaccination would be brought on by more information on the vaccine’s effectiveness (24%), on vaccine safety (22%) and regarding the existence of such a form of prevention (21%).

There were no statistically significant dependencies between these views compared to gender and specialization. There exists a difference of opinion on the co-financing of vaccinations by the work establishment between doctors among age groups (*p* = 0.024). Along with increasing age, a decreasing percentage agree with this proposition (Figure 6).

## 4. Discussion

Despite the recommendations of ACIP (Advisory Committee on Immunization Practices), the position of the WHO, recommendations of medical associations and provisions in the Protective Vaccinations Programme [1,6,12] the results of the conducted analysis indicate an insufficient degree to which the binding recommendations are implemented. The obtained data are concurrent with the results of other analyses conducted among the medical personnel and presented in the available literature [13,14,15,16,17,18,19,20,21,22].

Vaccination against hepatitis B (Hep B) was the most frequently used in the presented group, despite not being obligatory for these age groups in childhood. This vaccination has been obligatory in Poland since 1996. The people that had been vaccinated during childhood are only now finishing their medical studies. The hepatitis B infection is considered to be an occupational disease of medical personnel [1,4,5,12] and therefore obligatory; free vaccination is also provided to the medical employees of health care facilities and students in medical schools [5,6,12]. The protective equipment used by doctors is not a guarantee for safety, as according to research it can be often damaged or unreliable during operations or other medical procedures [23]. Only immunization provides adequate protection against infection of HBV in the case of a malfunction of gloves or accidental injury. It is surprising that as many as 26% of the respondents did not specify their hepatitis B vaccination. This demonstrates insufficient motivation of medical personnel toward vaccination, but also an insufficient verification of the degree of vaccination of personnel by employers in the health care industry. The hepatitis B vaccination is common in France, Netherlands, Latvia, Luxembourg, Malta, Slovakia and Slovenia—this vaccination is obligatory, similar to in Poland [1,7,8,11,12,13].

In the presented analysis, it was demonstrated that a majority of doctors (70%) vaccinated themselves against influenza, but only about half of the people declaring vaccination performed it on a regular basis. The age group more willing to vaccinate themselves were the doctors older than 51 years of age, which may be connected to more experience, but also to higher risk of post-flu complications for this age group. The percentage of respondents in the survey declaring this vaccination was significantly higher than in the national statistics (70% vs. 22.3%) [15,16,17,18] and was similar to the level in Germany. The results of studies on the degree of influenza vaccination of the hospital’s medical personnel indicate that this level is low in many countries. Germany has the highest number of vaccinated medical personnel in Europe (55%). A total of 72% of doctors and 45% of nurses are making use of the flu vaccine [14]. The advantages of this vaccine are very well-documented for both the medical personnel and for the patients [2,15,16,17,18,19,20]. A similar tendency was observed in other studies, including in a cross-sectional analysis conducted in the US [18]. The conducted studies show that personal recommendations and caring for a frail member of a family are the most motivating factors for regular vaccinations against the flu [17,18,19,20]. That could also be an explanation for the result of our study: the majority of surveyed family medicine specialists declared the annual flu vaccinations. It might be connected with their professional experiences both in the immunization of adults and in the treatment of the post-influenza complications among patients of their own age, which makes the recommendations more personal to them.

A total of 37% of the surveyed doctors had received booster doses of the tetanus vaccine. Most a significant part of them were probably a part of a post-exposure procedure after an injury.

Only 17% of the surveyed group of doctors had received a pertussis infection as adults. The booster doses of vaccination against this disease are recommended every 10 years. This applies in particular to persons working with infants and small children, due to the risk of transmission of the infection [24,25]. Pneumococcal vaccination was marked by 14% of the respondents. This vaccination is recommended in Poland for the 50 years and older age group, in particular in people with chronic pulmonary and cardiovascular diseases [7,8].

A decision to receive other recommended vaccinations was taken by approximately 20% of the surveyed persons. Some of these vaccinations were probably related to trips to other climate zones (e.g., yellow fever, cholera vaccines) and with active outdoor lifestyles (e.g., tick-borne encephalitis vaccine). However, there is no literature data on the frequency of such vaccinations among medical personnel and on their motivation.

In the presented study, most of respondents declared immunity against the measles virus. One in four doctors knew they were not immune or did not know whether they were immune. Approximately 15% had no knowledge about their degree of immunity to the measles virus. This may demonstrate a lack of knowledge of the current recommendations, as well as the fact that employers do not disseminate the recommendations nor encourage employees to follow them. The problem does not only concern doctors in Poland. In the last years, outbreaks of measles in Europe frequently were related to medical personnel that did not have appropriate immunization [26,27,28,29,30]. In Europe in 2017–2018, measles outbreaks appeared as a consequence of a decrease of post-vaccination immunity against this disease. Also, in Poland in 2014 a total of 144 cases of this illness were reed. The proposed preventive procedures in healthcare facilities include, among others, the assessment of the state of immunization of employees and additional vaccinations for potentially non-immune persons. Obligatory vaccinations of children against measles were introduced in Poland in 1975, and since 1991 they have a two-dose scheme, which provides long-term protection.

Half of the doctors participating in the study recommend their own families to vaccinate only girls against Human Papilloma Virus (HPV). Only 35% of surveyed doctors recommend this vaccination to children of both sexes, and one in five doctors did not recommend this vaccination to their families. A correlation of these recommendations with the doctor’s age and with the respondent’s specialization was demonstrated. It seems that younger doctors are more aware of current recommendations concerning the HPV vaccine and that most doctors working in specialties where they qualify young people for vaccinations do not hesitate to suggest these vaccinations to related children of both sexes.

The human papillomavirus (HPV) is considered to be the cause of cervical cancer in women and of other cancers in both sexes, including urogenital and also of the head and neck. Vaccination against the HPV virus is currently recommended to teenagers of both sexes in such countries as Austria, Belgium, Croatia, Czechia, Denmark, Finland, France, Germany, Hungary, Ireland, Italy, Netherlands, Norway [7,31,32]. In Bulgaria, Cyprus, Estonia, Iceland, Malta, Greece and Romania, recommendations and reimbursement apply to girls only. In Poland, recommendations apply to teenagers of both sexes; however, despite previous announcements, HPV vaccination was not included in the Protective Vaccinations Programme for 2022 as a free public vaccination, and it was only indicated as prioritized for financing as part of local government prevention programs. It is only from this year that it is obligatory and refunded to all Polish adolescents [7,31,32].

One in four respondents emphasized the lack of sufficient knowledge of the personnel on the possibility of protective vaccinations and on their safety and efficacy. This may indicate a significant gap in the education of doctors and other medical employees. An increase of vaccinations among health care employees according to many authors is related to education and to the provision of information on the benefits of preventive immunizations [18,19,20].

The survey was conducted during a phase of the COVID-19 pandemic when the vaccines were still under development. Now, as more than 2 billion COVID-19-vaccines have been given and we know from statistics and research that they are safe even for very specific groups—such as oncological patients [33] or young children [34]—the subject of vaccination is more widespread than before the pandemic, not only among medical staff. In Poland, immunization against COVID-19 is not included in the vaccination schedule, but it is strongly recommended and offered free of charge for all Polish citizens older than 6 months. It is mandatory for healthcare professionals.

It should be emphasized that the legal acts in force in Poland, particularly the Regulation of the Minister of Health of 22 April 2005 on the Harmful Biological Agents for the Health in the Work Environment and the Protection of the Health of Workers Working Exposed to These Factors [6,13], require the employer to, among other things, “order the execution of the exposure work to the harmful biological agent to the workers properly protected, including immunized with the available vaccines”. Unfortunately, the regulations do not establish in what manner and in what situations the employee is required to verify the immunization or to enforce the possible vaccination of the employees.

We have also not found unequivocal correlations between the decisions to vaccinate oneself and the sources of knowledge declared by the surveyed doctors. There was a statistical dependency between persons using the WHO website compared to other sources—both for measles vaccination (29% vs. 16%) and for pertussis booster vaccinations (28% vs. 14%). In the case of recommending HPV vaccination to their family, the proportions are reversed—it is recommended only by 10% of persons, using mainly the WHO website, whereas 20% of doctors who declare the use of other sources recommend it. Using the websites of medical associations was related to a higher tendency for pneumococcal vaccination (23% vs. 6% in people not using these websites).

The ambiguous results concerning the degree of vaccination and decisions concerning the vaccination of children in correlation to the use of social media are probably related to the fact that the most important factor in the use of this source is the respondent’s age. Additionally, social media is not only a source of knowledge but also of disinformation, inducing vaccination hesitancy even among health professionals, which is an increasing problem—not only in Poland [35,36,37,38,39,40].

## 5. Restrictions of the Study

Due to the relatively low number of conducted surveys, the material obtained from the analysis may not be treated as representative for the entire medical community in Poland.

## 6. Conclusions

The protection of medical employees against infection does not only serve their personal benefit, but also the safety of their patients and colleagues, and it reduces the risk of transmission of VPDs during the provision of health care services. The immunological resistance of personnel providing care to patients from risk groups, especially elderly patients, is of particular importance. The lack of knowledge and vaccination hesitancy among health care workers creates unacceptable danger in the healthcare facilities [21].

The necessity to avoid the spread of infectious agents and to protect the medical personnel forms the basis of many regulations, procedures and recommendations [1,4,6,7,41,42,43,44]. The employers of this sector should actively promote this form of prophylaxis to prevent contamination and minimize the risk of lawsuits for damages related to infections [13].

The propagation of protective vaccinations among medical employees requires an information campaign. It seems particularly important for doctors who are not dealing with vaccinations in their everyday professional practice.

It would be advantageous to amend the legal regulations so that vaccinations for the medical personnel are free or co-financed, easily accessible and possible to be monitored or verified by employers [45,46,47,48].

It is highly probable that promoting the idea of preventive vaccination among healthcare workers, in particular among doctors, and raising their competencies within this scope will also increase their engagement in recommending protective vaccinations to the patients [39].

## Figures and Tables

**Figure 1 ijerph-20-04153-f001:**
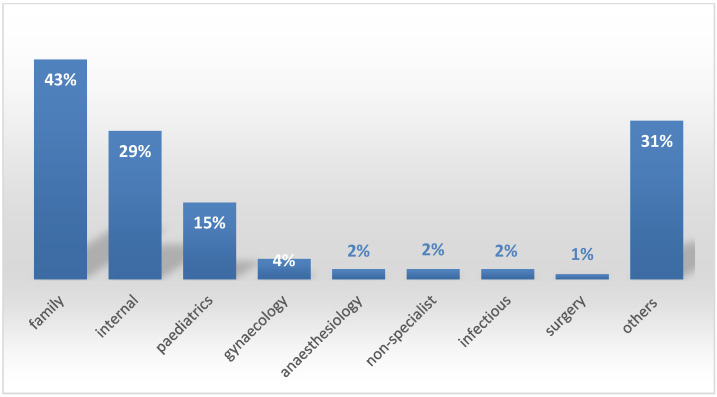
Specialties of the survey participants.

**Figure 2 ijerph-20-04153-f002:**
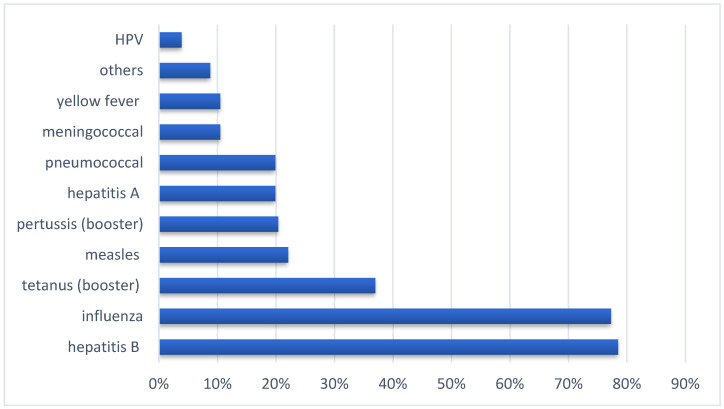
Additional vaccinations of doctors.

**Figure 3 ijerph-20-04153-f003:**
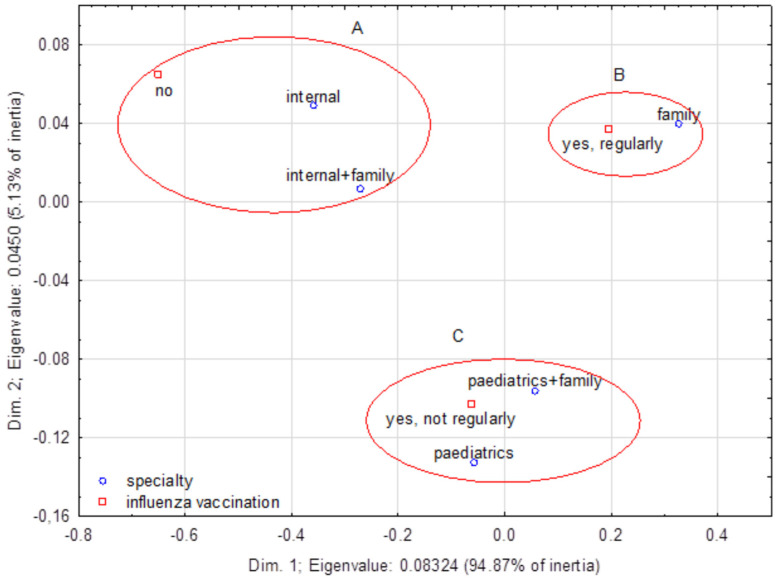
A correspondence map that demonstrates correlations between the respondent’s specialization and influenza vaccination.

**Figure 4 ijerph-20-04153-f004:**
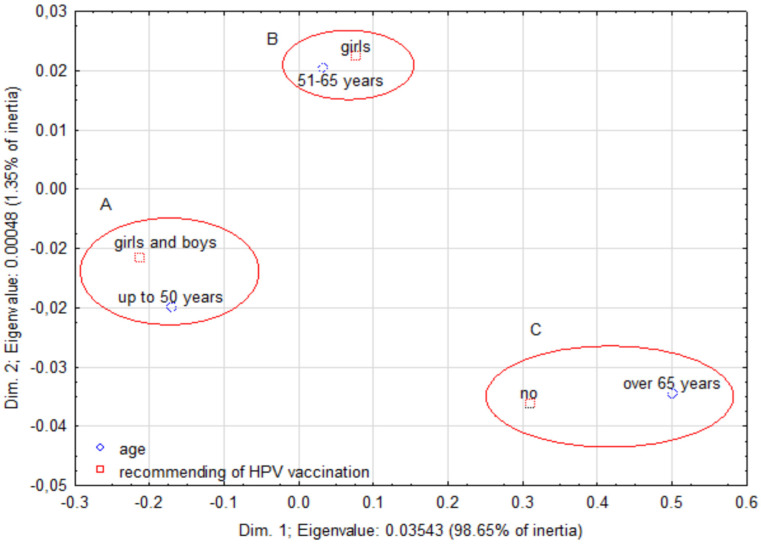
A correspondence map showing correlations between the doctors’ age and recommending of HPV vaccination to children of family and friends.

**Figure 5 ijerph-20-04153-f005:**
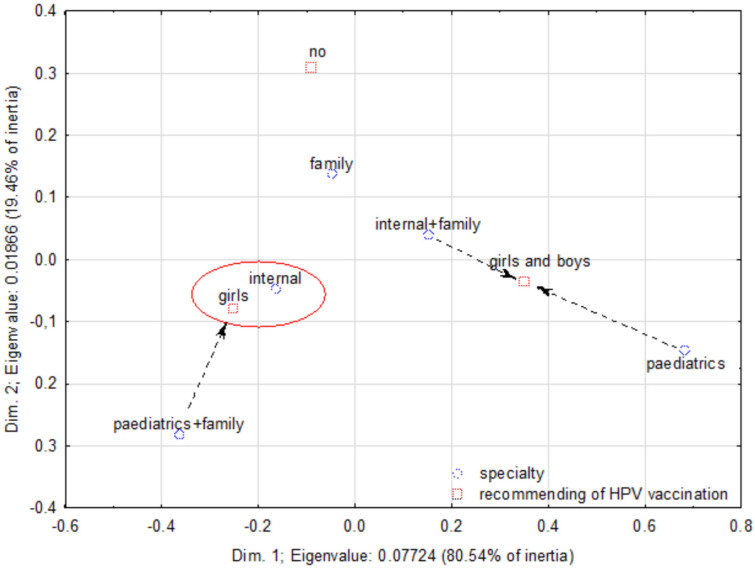
A correspondence map between the doctors’ specialty training and recommending of HPV vaccination to children of family and friends.

**Figure 6 ijerph-20-04153-f006:**
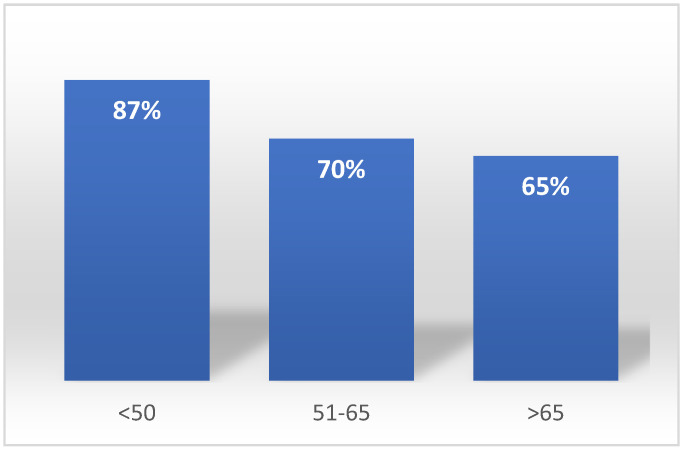
The respondent’s age and the opinion of the need to for the work establishment to co-finance vaccinations.

**Table 1 ijerph-20-04153-t001:** Detailed characteristics of respondents concerning vaccination.

Variable	Hepatitis B	Influenza	HPV	Tetanus	Hepatitis A	Measles	Mandatory Only	Pertussis	Pneumococcal	Meningococcal	Yellow Fever	Other
**Age (years)** **Female/Male** ***n* = 181**												
<50	46	50	3	24	14	17	10	8	10	8	7	9
50–65	83	77	3	38	20	21	8	29	23	10	9	6
>65	13	13	1	5	2	2	2	0	3	1	3	1
**Employment**												
Hospital *n* = 42	29	29	3	16	13	12	9	8	5	3	10	6
Infectious diseases or COVID hospital *n* = 11	6	8	0	3	3	1	1	1	4	2	0	1
Clinic *n* = 115	98	94	4	44	19	26	9	26	26	14	9	8
Private practice *n* = 11	9	8	0	4	1	1	1	2	1	0	0	1
**Specialty**												
Internal medicine *n* = 52	41	41	4	21	13	11	8	11	11	5	10	2
Infectious diseases *n* = 4	2	3	1	1	3	1	0	0	0	0	0	1
Anaesthesiologist *n* = 3	2	2	0	1	0	0	1	0	0	0	0	0
Surgery *n* = 2	1	1	1	1	1	1	1	1	0	0	1	1
Paediatrics *n* = 27	21	21	1	9	7	5	4	9	9	4	1	5
Family medicine *n* = 78	66	68	1	28	11	19	7	17	18	10	5	7
Gynaecology *n* = 7	5	3	0	1	1	1	2	1	1	1	1	1
Non-specialist *n* = 4	3	4	1	2	0	1	0	0	2	0	1	0
Other *n* = 57	49	41	0	22	11	7	5	7	5	3	7	7

## Data Availability

The data presented in this study are available on request from the corresponding author.

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
