# Peer review of "Protective Vaccination Used by Doctors for Prevention of Infections"

_ijerph, 2023, doi:10.3390/ijerph20054153_

Round 1

Reviewer 1 Report

Knowledge of the current status of preventive vaccination among healthcare workers is a valuable tool for the health administration.

1.     Can you exclusively mention the mandatory and optional vaccines of the geographical location in the introduction? It will provide further perspective on the information.

2.     Can you provide more details about the questionnaire used? Perhaps provide it as a supplement?

Author Response

Pleas see the attachment

Reviewer 2 Report

The methodology and methods of the paper appear to be sound. However, I have a reservation about the overall structure of the paper. The paper aims at studying the rate of vaccination by medical personnel; however, given the prevalence of Covid-19, it is surprising to find that the authors did not include vaccination against Covid-19 in their study. This becomes even more surprising when the authors themselves mention in the paper that SARS-CoV-2 represents an epidemiological condition, which seems to suggest that at least some measures need to be in place as a rationale for the paper.

So I would suggest that the authors include the rate of vaccination against Covi-19 in their study. And given the variety of Covid-19 vaccines, it would be recommendable to include these varieties in their study too. That would help bolster the authors' claim about the rate and the need for vaccination of medical personnel significantly.

Reviewer 3 Report

This study reports a survey results on immunization of 36 medical employees against Vaccine Preventable Diseases. 

The introduction can be completed with the following article on the importance of covid19 vaccination and its safety also in oncological patients (10.3390/vaccines10111887). 

Materials and methods are well presented. An example of the administered survey should be added.

Results are clear and interesting. The use of histograms and tables are helpful to have a clear pictures.

Discussion well written and the results are supported by the literature.

Conclusions should be more concise and driven by the results.

Round 2

Reviewer 2 Report

The authors have satisfactorily addressed the concern of the reviewer. I recommend publication now.